# OpenReview forum: "ML-Agent: Reinforcing LLM Agents for Autonomous Machine Learning Engineering"
_ICML.cc/2026/Conference — ICML 2026 regular_

### Official Review · Reviewer_2Evu · 2026-03-01

**Soundness:** 2
**Presentation:** 2
**Significance:** 3
**Originality:** 2
**Overall Recommendation:** 4
**Confidence:** 4

**Summary:**

This paper trains an ML engineering agent using reinforcement learning instead of relying only on prompting. The model is fine-tuned to iteratively improve ML code and performance across tasks.
The main contributions are:
1. Exploration-enriched fine-tuning to encourage diverse solution strategies before RL training.
2. Step-wise RL to accelerate experience collection by training on single steps rather than full rollouts.
3. An ML-specific reward module that handles improvements, errors, and edge cases in coding environments.

**Compliance With Llm Reviewing Policy:**

Affirmed.

**Final Justification:**

Thank you for the substantive follow-up. This response directly addresses my remaining concern on W2. The two empirical comparisons are exactly what I asked for.  The revised Related Work paragraph also clearly scopes the novelty without overclaiming. As I indicated I was open to, I am raising my score given that both the empirical grounding and the clearer positioning are now in place.

**Key Questions For Authors:**

Larger models incur higher per-step computational cost and memory overhead. Therefore, GPU-time comparisons across different model scales (Figure 3) may conflate architectural scale with algorithmic efficiency. Without normalization for parameter count or convergence speed, it is difficult to isolate whether the efficiency gains stem from the training method or simply from using a smaller backbone.

**Limitations:**

No, the paper does not sufficiently discuss limitations or potential societal risks.

**Strengths And Weaknesses:**

Strengths:
1. They train the agent with reinforcement learning instead of just relying on prompting.
2. They clearly explain the problems they are solving (exploration, training cost, reward design).
3. The step-wise RL idea helps reduce training time and computation cost.

Weaknesses:
1. The claim of being the first RL-trained LLM agent for autonomous ML engineering appears somewhat strong.
2. While ML-Agent shows strong and competitive results, the claim of “superior performance” appears overstated, particularly since comparisons to other learning-based agent approaches are missing. For instance, they should compare against models like CodeRL.
3. Table 1 highlights ML-Agent in bold despite some models achieving higher absolute scores. The authors should clarify whether bolding reflects best-in-class for model size, cost-efficiency, or another constrained setting.

---

> ### Author Rebuttal · Authors · 2026-03-31
>
> We thank reviewer for the evaluation and specific suggestions. We address each concern below.
>
> ### W1: "First RL-trained LLM agent for autonomous ML engineering" claim.
> We agree that claiming to be the "first" can appear overly broad and will remove it in the revised version, as our core contribution lies in the efficiency and generalization of the RL training framework on autonomous machine learning. To clarify the precise positioning and novel
> ty of our work, we distinguish our work from three paradigms:
>
> - RL for Code Generation (e.g., CodeRL, AlphaCode): Focuses on single-pass synthesis with static unit-test rewards, lacking the iterative, multi-step experimentation loop (observe, code, execute, refine) essential for ML engineering.
>
> - RL for Web/QA Agents (e.g., AgentQ, IPR): Operates in navigation/dialogue, lacking complex ML-specific code-execution feedback (e.g., runtime errors, continuous metrics).
>
> To our knowledge, utilizing a learning-based approach (specifically RL) to explore an agent's **cross-task generalization** and cost-efficiency in  autonomous machine learning remains unexplored. We will update the Related Work section to explicitly discuss them and better show our contributions.
>
> ### W2: Missing comparison with learning-based approaches (e.g., CodeRL)
> We thank the reviewer for rasing the comparison with learning-based approaches. We clarify that some previous work such as CodeRL or RLTF addresses **single-turn code generation from specifications, which differs fundamentally from autonomous machine learning**. Specifically, we highlight the key distinctions below:
>
> |Dimension|CodeRL / RLTF|Our Work(ML-Agent)|
> | - | - | - |
> |**Task**| Single-turn code generation (spec → code) | Multi-step autonomous ML engineering (observe → act → observe → act)|
> |**Interaction**|One-shot|15-step sequential MDP|
> |**Action**|Complete program|ML code editing + file operations|
> |**Feedback**| Unit tests (pass/fail)| Downstream ML task performance (e.g., accuracy, F1) across diverse ML tasks+execution errors|
> |**Evaluation**| APPS, MBPP (programming puzzles)| ML engineering benchmarks (e.g., MLE-bench with Kaggle-style ML tasks)|
> |**Generalization Goal**|Solve individual coding problems| Generalize across heterogeneous ML tasks (different data types, model families, problem domains)|
>
> Given these fundamental differences in problem formulation, a direct experimental comparison on the same benchmark may not lead to a fully fair or informative evaluation. However, we fully agree that discussing these works strengthens the paper's positioning. In the revised Related Work section, We will explicitly discuss CodeRL and other RL-for-code methods to clarify that they address a complementary but distinct problem: improving single-pass code generation quality, whereas our work targets the different challenge of training an agent that can autonomously conduct and generalize across end-to-end ML experiments.
>
> ### W3: Table 1 bolding criteria.
> Thank you for pointing this out. As stated in the Table 1 caption, **bolding denotes "the best open-source LLM-driven agent."** ML-Agent achieves the highest average performance gain (**16.40%**) across 3 held-in tasks and 10 held-out tasks among all evaluated methods, trailing only methods driven by GPT-5(which is closed-source).  To prevent confusion, the revised table will bold the best open-source score per column (task). We will also underline the best closed-source score in each column for clear comparison.
>
> ### Q1: GPU-time comparison are across different model scales.
> We want to clarify that Figure 3 does not compare different model scales. Both ML-Agent (Step-wise RL) and standard PPO (Episode-wise RL) **use the same 7B backbone and identical hyperparameters**, so model size, per-step computational cost, and memory overhead are fully controlled. The only difference is the training paradigm—single-step updates versus full-trajectory rollouts. Therefore, the efficiency gains can be cleanly attributed to the algorithmic advantage of step-wise sampling, which decouples state collection from online rollout and removes the bottleneck of waiting for complete 15-step trajectories.
>
> ### L1: Limitations.
> Due to resource limitations, the LLM-based agent is trained on only 9 ML tasks. A more diverse set of tasks is necessary to understand its scalability.  Additionally, for significantly larger repositories, exploration requires more iterations. This drastically reduces episode-wise training efficiency and inflates inference costs for proprietary models, whereas our step-wise method remains highly efficient and cost-effective. However, extremely long trajectories may exceed current LLM context windows. Therefore, context management and compression for complex autonomous ML tasks is a possible future direction. We will include this discussion in the revision.

---

> > ### Author Rebuttal · Reviewer_2Evu · 2026-04-02
> >
> > Thanks for the detailed rebuttal. Most of my concerns are addressed. The clarification on W3 and the commitment to fix the bolding is appreciated, and the explanation for Q1, that both RL variants use the same 7B backbone, directly resolves my concern about conflating model scale with algorithmic efficiency.
> > On W2, I appreciate the comparison table distinguishing the task setting from CodeRL and RLTF. The differences are real, but the response is qualitative. The absence of any empirical comparison, even a partial one, still leaves the "superior performance" claim difficult to evaluate rigorously. The argument that a direct comparison wouldn't be fair is understandable, but it also makes it harder to assess the true contribution relative to prior learning-based approaches.
> > W1 is resolved by the authors' agreement to remove the claim.
> > I'm willing to raise my score slightly given the overall responsiveness of the rebuttal, but the missing empirical grounding in W2 and the low soundness and originality scores remain concerns I cannot fully set aside without seeing a revision. Maintaining my score of 3 for now, but open to reconsideration if the revised related work section makes the positioning significantly clearer.

---

> > > ### Author Response · Authors · 2026-04-07
> > >
> > > **Dear Reviewer 2Evu,**
> > >
> > > Thank you for your continued engagement and follow-up. To address the need for stronger empirical grounding on the distinction from prior RL-for-code methods, we added **two complementary empirical comparisons** and **revised the Related Work section**.
> > > ## 1. Empirical Clarification on Learning-Based Comparisons
> > > We add two comparisons to respond more rigorously to W2.
> > >
> > > First, we test whether our agentic-ML training framework(trained on **9 autonomous ML tasks**) could transfer to standard **single-turn code generation**. Following CodeRL, we compare the same **Qwen2.5-7B** backbone on **APPS** and **MBPP** before and after agentic ML training. As shown in **Table R4.1**, our training consistently improves both benchmarks, suggesting the framework does not simply overfit to autonomous ML, but also learns capabilities that transfer to one-shot code generation.
> > >
> > > **Table R4.1 Comparison on code generation benchmarks.**
> > > |Benchmark|APPS(Intro) pass@1|APPS(Inter) pass@1|APPS(Comp) pass@1|APPS(All) pass@1|MBPP pass@80|
> > > |-|-|-|-|-|-|
> > > |**CodeT5**|6.60|1.03|0.30|2.00|46.8|
> > > |**+CodeRL** *(trained on APPS)*|7.08(+0.48)|1.86(+0.83)|0.75(+0.45)|2.69(+0.69)|63.0(+16.2)|
> > > |**Qwen2.5-7B**|27.30|8.20|2.60|10.90|76.9|
> > > |**+ours agentic ML training** *(trained on 9 ML tasks)*|35.40(+8.10)|11.70(+3.50)|3.70(+1.10)|14.84(+3.94)|82.6(+5.7)|
> > >
> > > Second, since APPS/MBPP do not capture the **multi-step, interactive** nature of autonomous ML engineering, we further adapt the **RL-for-code paradigm inspired by CodeRL** to our setting and evaluate it on held-out tasks, using the **same Qwen2.5-7B backbone**. To stay faithful to CodeRL, this baseline uses standard SFT, episode-wise REINFORCE, and a CodeRL-style execution-oriented reward. Results are in **Table R4.2**.
> > >
> > > **Table R4.2 Comparison on agentic ML tasks.**
> > > |Method|SFT paradigm|RL paradigm|Reward design|Avg. Perf. Gain(Held-out)|
> > > |-|-|-|-|-|
> > > |**CodeRL-style Baseline**|Standard SFT|Episode-wise RL using REINFORCE|Intermediate return by critic as error predictor|1.56%|
> > > |**ML-Agent(Ours)**|Exploration-enriched fine-tuning|Step-wise RL|Agentic ML-specific reward|**15.91%**|
> > >
> > > We do not view this as diminishing CodeRL in its intended domain. Instead, it shows that a paradigm designed for **single-turn program synthesis with execution-based correctness signals** does not transfer directly to **long-horizon autonomous ML engineering**, where agents must iteratively inspect feedback, revise ML code, execute pipelines, and optimize against diverse signals including metric changes, failures, and ML-specific corner cases. This strengthens the **soundness** of our claim: the gain comes not merely from using RL, but from tailoring the full training pipeline to **autonomous machine learning**.
> > > ## 2. Clarifying the Scope of Originality
> > > We thank the reviewer for raising the originality concern. We do not claim novelty in the RL optimizer itself; our contribution is to make RL effective for **autonomous machine learning**, a long-horizon, high-latency code-execution setting with varied ML feedback. The novelty lies not in any isolated component, but in integrated learning framework for agentic ML, enabling **cross-task generalization**, **efficient training**, and **substantially lower computation cost**. This allows a **7B** agent to learn from experience and remain competitive with much larger prompt-based methods.
> > > ## 3. Revised Related Work
> > > To clarify the positioning, we will add the following paragraph to Related Work.
> > >
> > > > **Reinforcement Learning for Code Generation.** Recent work shows that RL can improve LLM coding ability. Pioneering work such as CodeRL[1] trains a code model using compiler and unit-test feedback, and introduces a critic to provide intermediate value estimates during single-turn program generation. These methods have demonstrated the effectiveness of RL for code generation, particularly in single-turn program synthesis settings evaluated on benchmarks such as APPS and MBPP. In contrast, autonomous ML engineering requires multi-step interaction with an experimental environment: agent must repeatedly inspect feedback, edit ML code, execute pipelines, and refine its strategy over a long horizon. Feedback is also more varied, including continuous downstream ML metrics (e.g., ACC, log-loss), execution failures, and ML-specific corner cases. Our framework is designed for this setting: step-wise RL improves training efficiency by avoiding full-trajectory rollouts, while the ML-specific reward module converts heterogeneous experimental feedback into a unified training signal.
> > >
> > > We believe that these added experiments and the clearer positioning address the remaining concerns on soundness and originality, and hope this clarification may merit reconsideration of the overall score.
> > >
> > > [1] H. Le, Y. Wang, A. D. Gotmare, S. Savarese, and S. C. H. Hoi, "CodeRL: Mastering Code Generation through Pretrained Models and Deep Reinforcement Learning," arXiv:2207.01780, 2022.

---

### Official Review · Reviewer_Bd5k · 2026-03-08

**Soundness:** 3
**Presentation:** 3
**Significance:** 2
**Originality:** 2
**Overall Recommendation:** 4
**Confidence:** 3

**Summary:**

This paper proposes ML-Agent, an RL framework for training agents on machine learning tasks. This framework comprises three components: exploration-enriched fine-tuning, step-wise RL and agentic ML-specific re-
ward module. Results indicate that this framework effectively boost performance over base model, and even competitive against proprietary models, and this performance also generalizes to held-out tasks. Ablation studies demonstrate the effectiveness of three modules in the proposed framework.

**Compliance With Llm Reviewing Policy:**

Affirmed.

**Final Justification:**

This paper presents ML-Agent, a step-wise RL framework for training agents on science tasks and aims to address training efficiency issue in the machine learning domain. Their proposed method achieves better GPU time utilization and also yields strong performance on both held-in and held-out tasks. Additionally this paper provides thorough ablation studies that offer clear evidence of each module's effectiveness and also valuable technical insights. Although there remains concern whether and how their method can be adopted for more complex AI Scientist tasks, their framework's solid implementation and empirical success strongly support its contribution and application value. This paper should be recommended for acceptance.

**Key Questions For Authors:**

1. It'd be helpful if authors could elaborate on the first point mentioned in Weakness, especially how their method could be adapted to complex agentic scenarios which involves planning, reasoning and more diverse interactions.
2. More discussion of exploration-enriched fine-tuning is needed ragarding its role and the comparison against RL contribution.

**Limitations:**

Whether the proposed framework can be applicable to more complex agentic ML tasks should be discussed.

**Strengths And Weaknesses:**

## Strengths:

The proposed framework effectively addresses the challenge of RL training efficiency in the ML domain. As is shown in Figure 3, this framework achieves significantly better GPU time utilization compared to episode-wise RL, providing a practical alternative for training RL on time-consuming tasks.


## Weaknesses:

1. This paper adopts step-wise RL paradigm to solve the challenge of training efficiency. Though this is proved effective, it introduces a partially offline RL setting. In this method, all states are pre-collected with stronger models, and the trained agent cannot explore in the environment. This leads to a heavy dependency on the quality and diversity of pre-collected states. More critically, real ML tasks is usually more complex, involving proposing ideas, exploring codebase, editing files, and iterating to optimize performance. In the current training paradigm, the step-wise paradigm and reward design makes it hard for the agent to learn how to conduct planning and take consecutive actions, which is necessary in more complex scenarios such as those with a large codebase.
2. Another thing to notice is the role of exploration-enriched fine-tuning. As is shown in Table 2 and Figure 4, removing this module leads to more significant performance degradation compared with other two modules. The effect of this module might be understated, and based on the result, a conclusion should be that SFT actually contributes more to the final result.

---

> ### Author Rebuttal · Authors · 2026-03-31
>
> We thank the reviewer for the constructive comments.
> ## W1.1: Dependency on Pre-Collected States and Lack of Exploration
> **1. The agent actively explores in the environment:** Rolling out from pre-collected expert states does not prevent exploration. During RL, the agent is **not** forced to imitate the expert actions. Instead, it freely proposes new actions from these states and adjusts its policy using environment rewards.
>
> **2. Robustness to the quality of pre-collected states:** We do rely on a stronger model to collect initial states(as weak models fail to generate valid/formatted actions), but our method does not depend on the quality or optimality of the expert.
> * The expert(GPT-4o-mini) used to collect the data is capable but not top-tier: its average performance gain is 3.11%, while our ML-Agent achieves 16.40%.
> * Expert trajectories only expose valid regions of the state space; they do not define the optimum. Our agentic-ML reward penalizes failures, and RL drives the policy away from suboptimal behaviors.
>
> **3. Dependency on state diversity:** We fully agree that diversity is critical. This is exactly what exploration-enriched fine-tuning is designed for. Our ablations show that standard SFT yields low action diversity, severely limits later RL generalization. Our tailored fine-tuning broadens the state pool and enables effective RL exploration.
> ## W1.2&Q1&L1: Applicability to Complex Agentic Scenarios
> **1. The current environment is inherently complex:** As detailed in **Appendix Tables 5 and 7**, our environment already requires the agent to explore repo-level codebases, understand and edit files, and iterative optimization. The task formulation also explicitly requires reasoning, high-level research plan, and executing consecutive multi-step actions.
>
> **2. Step-wise RL successfully enables planning and consecutive actions:** Although RL rollout is single-step, the **state representation** contains rich history(multi-step feedback and previous actions). Moreover, each step in our framework includes a full think-then-react cycle and can implement compound ML coding strategies. To prove this, we analyzed rollout trajectories on 10 held-out tasks before and after applying our step-wise RL with the ML-specific reward(Table R3.2). The results show clear gains in planning-related behaviors:
>
> **Table R3.2:** Behavioral metrics before and after Step-Wise RL on 10 held-out tasks.
> |Metric|Before(%)|After(%)|Δ|
> |-|-|-|-|
> |Error Execution Rate|14.6|6.5|-55%|
> |Backtracking Actions|2.4|2.9|+17%|
> |Hypothesis-driven action Rate|1.8|2.6|+44%|
> |Composite ML strategies rate|16.0|32.5|+103%|
>
> **Appendix C.5** further shows case studies where the trained agent generates diverse, reasoning-driven action sequences and automatically backtracks when metrics fail to improve.
>
> **3. Whether the proposed framework can be applicable to more complex agentic ML tasks should be discussed.:** We agree that larger repositories require more exploration steps, making episode-wise training less efficient and proprietary-model inference more costly. Our step-wise method remains more efficient, but very long trajectories may exceed current LLM context window. We will add this limitation and discuss context management/compression in the revision.
> ## W2&Q2: The Role of Exploration-Enriched fine-tuning vs. Step-Wise RL
> We first clarify a detail in Table 2: the 'w/o step-wise RL' actually represents SFT + episode-wise RL, not pure SFT. As shown in Figure 3, a purely exploration-enriched SFT model achieves 8.35% on held-in tasks but only **0.41%** on held-out tasks.
>
> **Why removing SFT causes performance collapse:** A base model without SFT lacks basic instruction-following and format compliance. Without the ability to generate valid actions, the agent cannot interact with the environment to receive meaningful reward signals, so RL training collapses.
>
> **SFT and RL play complementary roles:** SFT is necessary but not sufficient: it provides basic autonomous ML capabilities and format compliance, but without RL, the agent does not generalize. As shown in Table R3.3, SFT achieves 8.35% on held-in tasks but only 0.41% on held-out tasks; more SFT only overfit to the held-in(10.55%) and does not yield generalization(-14.47% in held-out); while the full ML-Agent reaches 18.01% and 15.91% respectively. This shows that SFT mainly supplies valid and diverse actions, while step-wise RL is the key driver of reward-guided refinement, efficiency, and cross-task generalization.
>
> **Table R3.3:** Breakdown of SFT vs RL contributions.
> |Method|Held-in(%)|Held-out(%)|
> |-|-|-|
> |Expert(GPT-4o-mini)|6.17|2.19|
> |Exploration-enriched fine-tuning|8.35|0.41|
> |+Exploration-enriched Fine-tuning|10.55|-14.47|
> |+Step-Wise RL(ML-Agent)|**18.01**|**15.91**|
> Since our core claim is **strong cross-task generalization at much lower computational cost**, RL is the decisive contributor to the main findings. We will make this distinction clearer in the revision.

---

> > ### Author Rebuttal · Reviewer_Bd5k · 2026-04-02
> >
> > Thanks for the authors' response and valuable analyses. My concerns have been largely resolved, and Table R3.3 is a good supplement in illustrating the separate role of SFT and RL. One remaining point is the complex scenarios, where I'd like to mention those where outcome changes may result from several consecutive steps. I wonder how step-wise RL could solve such cases and how the reward should be designed to properly account for them. It'd be helpful if authors could further discuss on this. I'd like to raise my rating to 4.

---

> > > ### Author Response · Authors · 2026-04-04
> > >
> > > Dear Reviewer Bd5k,
> > >
> > > We deeply appreciate your constructive and timely feedback, and we are glad that Table R3.3 helped resolve your initial concerns. We also sincerely appreciate your willingness to raise the score. Regarding your follow-up question on complex scenarios requiring consecutive steps, we are glad to provide detailed specifics below.
> > >
> > > **Q1: Outcome changes may result from several consecutive steps in complex scenarios. How could step-wise RL solve such cases, and how should the reward be designed to properly account for them?**
> > >
> > >
> > > This is a fundamental challenge in RL, often manifesting as the credit assignment problem.
> > > In outcome-reward-based training methods, agents usually receive a sparse **Outcome Reward** only after completing a long chain of consecutive steps, making it extremely difficult to distribute that reward to the intermediate actions that contributed to the final outcome. In contrast, our framework utilizes a dense **Process Reward** combined with the decoupled state sampling of Step-Wise RL. Specifically:
> > >
> > > ## **1.Step-wise RL improves sampling efficiency without compromising reward density.**
> > > By decoupling the trajectory into a pre-collected state pool, our state pool contains the intermediate states of **every step** within a consecutive sequence.
> > >
> > > - In standard **episode-wise RL**, getting the reward $R(s_{t+1}, a_{t+1})$ requires **sequentially** waiting for the environment to execute $a_t$, transition to $s_{t+1}$ and return $R(s_t, a_t)$, while $R(s_t, a_t)$ itself depends on the execution of even earlier steps, which is not efficient.
> > >
> > > - In contrast, our **step-wise RL** samples $s_{t+1}$ directly and independently from the pre-collected expert state pool and does not depend on the agent's current execution at $s_t$. The agent rolls out atomic actions from these various breakpoints, meaning **every step in a consecutive chain** receives reward feedback ($R(s_{0}, a_{0})$,...,$R(s_t, a_t)$,$R(s_{t+1}, a_{t+1})$) based on our Agentic ML-specific reward module **in parallel**.
> > >
> > > In short, we eliminate the sequential rollout bottleneck and improve the training efficiency while **every single step still receives its own reward and the reward is still dense**.
> > >
> > >
> > > ## **2. A Concrete Example of How Rewards are Assigned to Consecutive Steps.**
> > > Consider a complex scenario requiring **$t$ consecutive steps**.
> > >
> > > | Step | Action            | Details                           | Reward                      |
> > > | :--- | :---------------- | :-------------------------------- | :----------------------------------- |
> > > | $0$    | `List Files`      | Explore codebase                  | $0$ (Valid neutral)                  |
> > > | $1$    | `Understand File` | Understand certain aspects| $0$ (Valid neutral)|
> > > | $2$    | `Edit Script`     | Execute and get **bugs**          | $-1$ (Error penalty)                 |
> > > | ...    | ...     | ...                       | ...|
> > > | $t-3$    | `Edit Script`     | Propose the Data aug. idea -> **Worse** results    | Negative $\Delta$ (Performance drop) |
> > > | $t-2$    | `Edit Script`     | Propose the Change LR idea-> **Better** results   | Positive $\Delta$ (Performance gain) |
> > > | $t-1$    | `Edit Script`     | Propose the Add CNN layers idea-> **Much better** results | Positive $\Delta$ (Performance gain) |
> > >
> > > During our Step-wise RL training, the agent is **not** forced to stumble through steps $0$ to $t-2$ just to learn how to act at step $t-1$. Instead, our Step-wise RL independently and equally samples state $s_0$ and learns how to act $a_0$, ..., samples state $s_{t-2}$ (which already contains the history of steps $1,2,...,t-3$ ) and learns how to act $a_{t-2}$. The agentic-ML specific reward is a process reward: It precisely measures the quality of the action taken *at that specific intermediate state* (e.g., $0$ for valid preparation, $-1$ for syntax errors, and scaled positive/negative rewards for performance changes), and the agent explicitly learns what constitutes a "good" or "bad" action at every step.
> > >
> > >
> > > In summary, rather than struggling with sparse outcome rewards at the end of a long consecutive steps, our framework decomposes the credit assignment problem. By independently sampling historical states and applying dense process rewards at each step, the agent explicitly learns robust strategies for planning, debugging, and optimizing at each given phase of a complex ML lifecycle.

---

### Official Review · Reviewer_CP5M · 2026-03-12

**Soundness:** 3
**Presentation:** 3
**Significance:** 3
**Originality:** 3
**Overall Recommendation:** 4
**Confidence:** 3

**Summary:**

The paper proposes ML-Agent, a framework that trains a 7B LLM agent via reinforcement learning to autonomously perform ML engineering tasks. The training pipeline has three stages: exploration-enriched SFT on GPT-4o-mini trajectories, step-wise RL that samples states from the expert distribution and evaluates only atomic actions (avoiding costly full-trajectory rollouts), and a task-specific reward module with normalized performance gains and corner-case penalties. Trained on 9 tasks from MLE-bench, ML-Agent achieves strong held-in performance (avg 72.89%) and competitive held-out results (avg 16.40%) at roughly 10-100x lower inference cost than prompt-based methods with proprietary LLMs (Fig. 2).

**Compliance With Llm Reviewing Policy:**

Affirmed.

**Final Justification:**

The rebuttal fully addresses my concerns. I maintain my positive score.

**Key Questions For Authors:**

Q1. Table 2 shows that removing SFT collapses performance while removing step-wise RL retains most of the gains. Can you provide a more detailed analysis of what RL learns beyond SFT, e.g., action patterns that appear after RL but not in the expert trajectories? This would clarify whether RL is discovering new strategies or mainly refining the SFT policy, and could change my assessment of the core contribution.

Q2. On held-out tasks, ML-Agent scores near zero on jigsaw (0.01) and tabular (0.20). What characterizes the held-out tasks where the agent fails? Is this a coverage issue (no similar training tasks) or a fundamental limitation of the 7B model? If coverage explains most failures, the path to improvement is clear; if not, the generalization claim needs revision.

Q3. Fig. 5 shows monotonic improvement from 3 to 9 training tasks. Have you estimated how many tasks would be needed to close the held-in/held-out gap, or whether performance plateaus? This has direct implications for the practical scalability of the approach.

Q4. How is $m_{\text{best}}$ (Eq. 6) determined for each task? Is it from leaderboard scores, the expert trajectories themselves, or manual specification? The reward scale depends on this value, and inconsistencies across tasks could bias training.

**Limitations:**

Partially. The authors discuss computational costs and evaluation scope, but do not address the held-in/held-out performance gap, the dominance of SFT over RL in the ablation, or the overstated comparison with GPT-5.

**Strengths And Weaknesses:**

Strengths:
- The step-wise RL formulation is well-motivated. Full-trajectory rollouts in agentic ML are prohibitively slow, and replacing them with atomic-action evaluation from an expert state distribution (Eq. 3-4) is a practical and clean solution. The engineering rationale is clearly presented.
- The reward design covers common failure modes (invalid/error = -1, corner = 0, normalized metric improvements), making training more controllable than a sparse final-score signal. Table 3 confirms all three components contribute.
- The cost-performance trade-off is the paper's strongest empirical result. Fig. 2 shows ML-Agent achieves ~15% avg gain on held-out tasks at ~$0.003 per trajectory, roughly 10-100x cheaper than prompt-based alternatives at comparable or better performance.

Weaknesses:
- The generalization gap between held-in and held-out tasks is large and insufficiently discussed. Table 1 shows 72.89% avg on held-in vs 16.40% on held-out, with several held-out tasks near zero (jigsaw 0.01, tabular 0.20). The claim of being "comparable to DeepSeek-R1 and GPT-5" is also overstated: on held-out tasks, ML-Agent (16.40) trails GPT-5 under all three scaffolds (MLAB 18.14, AIDE 20.95, ML-Master 19.12). The paper should present these comparisons more candidly.
- Table 2 reveals a strong asymmetry in the ablation: removing exploration-enriched SFT collapses performance entirely (-0.66 / -6.20), while removing step-wise RL retains meaningful gains (10.71 / 2.86). This suggests the system's capability is primarily grounded in expert trajectory distillation, with RL providing incremental refinement. The paper frames RL as the core contribution, but the evidence points to SFT initialization doing most of the heavy lifting.
- The step-wise RL replaces the online policy distribution $d^{\pi_\theta}$ with a fixed expert distribution $d^{\pi_e}$ for state sampling (Eq. 4). The training curve in Fig. 3 shows continuous improvement, suggesting the mismatch is not catastrophic in practice, but the paper provides no analysis of why this works, e.g., whether $\pi_\theta$ stays close to $\pi_e$ or the short training horizon simply prevents significant divergence.

---

> ### Author Rebuttal · Authors · 2026-03-31
>
> We thank the reviewer for the thorough suggestions.
> ### W1.1 & L1.1: Generalization gap
> We clarify a likely misreading: "72.89%" is the score on one held-out task(whale categorization), not the held-in average. The actual averages in Table 2 are **18.01%** on held-in and **15.91%** on held-out tasks, a gap of only **2.1** points.
>
> Thus, the gap is much smaller than suggested. The variance mainly reflects task difficulty rather than systematic overfitting. Notably, the near-zero tasks are equally difficult for all methods: on jigsaw, 11/14 method-model pairs score≤0.04, and even the best result is only 0.35. We will clarify this held-in/held-out comparison and the per-task difficulty in the revision.
> ### W1.2 & L1.3: Overstated comparison with GPT-5
> We note that 16.40/18.14/20.95/19.12 are **overall averages across all 13 tasks**, not held-out averages. Since our main claim is **cross-task generalization at much lower cost**, the more relevant comparison is the 10 held-out tasks together with per-trajectory cost(Figure 2). We tabulate some data below:
>
> |Method|Model|Held-out(%)|Cost/traj.($)|
> |-|-|-|-|
> |MLAB|GPT-5|14.95|$0.2025|
> |AIDE|GPT-5|16.50|$0.1746|
> |ML-Master|GPT-5|14.41|$0.0866|
> |**ML-Agent**||15.91|$0.0047|
>
> On held-out tasks, ML-Agent(15.91%) **outperforms** MLAB+GPT-5(14.95%) and ML-Master+GPT5(14.41%), and is close to AIDE+GPT-5(only 0.58pp gap), while being 18–43× cheaper. It indicates **ML-Agent achieves results comparable to agents using the most advanced LLMs but at significantly lower cost**. We will revise the paper to state more clearly that AIDE+GPT-5 achieves slightly higher held-out performance.
> ### W2&Q1&L1.2: SFT dominates RL in ablation and what RL learns beyond SFT.
> We first clarify that in Table 2: the "w/o step-wise RL" is not pure SFT; it is SFT + episode-wise RL. The pure exploration-enriched SFT achieves **8.35%** on held-in but only **0.41%** on held-out.
> **Why removing SFT causes collapse:** A base model without SFT lacks basic instruction-following and format compliance. If it cannot produce valid actions, it cannot interact with the environment reliably or receive meaningful reward, so RL training collapses.
> **SFT and RL are complementary:** SFT is necessary but not sufficient: it provides basic autonomous-ML ability and format compliance, but not generalization; More SFT improves fitting to seen tasks but not unseen ones; in Table below, adding more SFT raises held-in performance but reduces held-out performance to -14.47%. RL enables active exploration and learning from feedback, lifting held-out performance from 0.41% to 15.91%.
> |Method|Held-in(%)|Held-out(%)|
> |-|-|-|
> |Exploration-enriched fine-tuning|8.35|0.41|
> |+Exploration-enriched Fine-tuning|10.55|-14.47|
> |+Step-Wise RL (ML-Agent)|**18.01**|**15.91**|
> Since our main claim is **strong cross-task generalization at a much lower computational cost**, RL is the decisive contributor to our main finding. We will clarify the distinct roles of SFT and RL in the revision.
> ### W3.2: Why expert state distribution works
> The SFT-initialized policy is close to expert policy that generated the state pool and PPO’s KL penalty further limits drift. This likely keeps the distribution mismatch manageable in practice.
> ### Q1: What RL learns beyond SFT?
> We quantify behavioral changes on 10 held-out tasks before and after step-wise RL(%):
> |Metric|Before|After|Δ|
> |-|-|-|-|
> |Error Execution Rate|14.6|6.5|-55%|
> |Backtracking Actions|2.4|2.9|+17%|
> |Hypothesis-driven action Rate|1.8|2.6|+44%|
> |Composite ML strategies rate|16.0|32.5|+103%|
> RL reduces execution errors(-55%) while increasing backtracking, hypothesis testing, and multi-step strategy composition(+103%). This suggests RL learns more complex problem-solving behaviors beyond imitation. We will add this analysis in the revision.
> ### Q2: What characterizes failure held-out tasks?
> The two highlighted failures are not specific to ML-Agent: **jigsaw**: Most methods score near zero, suggesting that the task likely requires specialized toxicity-detection knowledge beyond what current general ML agents provide. **tabular**: GPT-5 reaches 0.23% and ML-Agent 0.20%. The initial script is already near-optimal(96% Acc.), leaving limited headroom.
>
> More broadly, held-out task performance depends on training-task similarity, improvement headroom, and metric sensitivity. This suggests broader task coverage is the clearest path to improvement.
> ### Q3: Task scaling and closing the held-in/held-out gap
> We observe no plateau up to 9 tasks: held-out improves from ~0%(0 tasks), ~3%(3 tasks), ~6%(6 tasks) to ~16%(9 tasks). This suggests that task *diversity* (not just count) is the key driver. However, scaling to more tasks is currently expensive in our setup, so we cannot yet estimate the plateau point reliably.
> ### Q4: How is $m_{best}$ determined?
> $m_{best}$ in Eq. 6 is the **top public Kaggle leaderboard score** for each task, providing a consistent and externally validated reference point.

---

> > ### Author Rebuttal · Reviewer_CP5M · 2026-04-03
> >
> > The rebuttal fully addresses my concerns. I maintain my positive score.

---

> > > ### Author Response · Authors · 2026-04-04
> > >
> > > Dear Reviewer CP5M,
> > >
> > > We sincerely thank you for your time and your acknowledgement to our rebuttal. Thanks again for your constructive and insightful feedback.
> > >
> > > Best regards,
> > >
> > > Authors

---

### Official Review · Reviewer_iha9 · 2026-03-14

**Soundness:** 3
**Presentation:** 3
**Significance:** 3
**Originality:** 2
**Overall Recommendation:** 4
**Confidence:** 3

**Summary:**

This paper presents ML-Agent, an RL-trained agent for autonomous ML engineering. The method combines exploration-enriched SFT, step-wise RL from expert intermediate states, and an ML-specific reward module that maps different environment execution feedback into RL rewards. Experiments with a 7B backbone show clear gains over prompt-based open-source baselines and performance approaching some much stronger closed-source agents at lower cost.

**Compliance With Llm Reviewing Policy:**

Affirmed.

**Final Justification:**

My concerns have been addressed. Therefore, I keep my original score.

**Key Questions For Authors:**

1. Can the authors quantify more precisely what the proposed SFT stage improves relative to standard SFT? For example, does it primarily improve action diversity, trajectory coverage, or early RL success rate?
2. How well does the method generalize beyond the benchmark family used in the paper?

**Limitations:**

A more concrete failure analysis would strengthen the paper. What types of tasks does the agent fail most often? Is the main issue long-horizon planning? Such an analysis would make the limitations significantly more informative.

**Strengths And Weaknesses:**

Strengths:
- The paper is clearly written and easy to follow.
- Autonomous ML engineering is a valuable direction of both scientific and practical interest.
- The individual ingredients (i.e., SFT, PPO, and reward shaping) are not novel in isolation, but adapting them to the specific constraints of expensive ML environments and demonstrating that this combination works well is a meaningful contribution.
- The paper presents a practically relevant result: with appropriate training, a 7B model can approach the performance of some significantly stronger proprietary agents on ML engineering tasks, while operating at much lower cost.
- The experiments also include a relatively complete ablation study.

Weaknesses:
- Although the paper distinguishes between held-in and held-out tasks, both training and testing tasks appear to come from closely related benchmark ecosystems, with similar environment interfaces, task styles, and code organization.
- The reward design is intuitive, but the taxonomy of error types could be specified more systematically. For example, it would help to clarify what qualifies as a corner case versus a definite failure. Without more precise definitions, it is possible that the reward shaping is heavily benchmark- or environment-specific.
- Step-wise RL depends on the expert state distribution. Because training mainly occurs on intermediate states induced by expert trajectories, the learned policy may be better characterized as making improvements near expert-reachable states rather than learning long-horizon planning and recovery from errors.

---

> ### Author Rebuttal · Authors · 2026-03-31
>
> ### **W1 & Q2: Similar benchmark ecosystems and generalization beyond the benchmark family**
>
> Thank you for raising this concern. We agree that related benchmark ecosystems may introduce potential over-specialization.
>
> **On environment and code standardization.**
> The shared interface (file operations + script execution) follows Kaggle-style workflows and ensures controlled comparison (same tools and time budget). It is minimal and does not constrain agent behavior (Appendix). Thus, generalization arises from diversity in tasks, data, and evaluation metrics rather than interface differences.
>
> **On held-out generalization within the benchmark family.**
> Held-out tasks are heterogeneous and test adaptation of *ML strategies* (model selection, preprocessing, optimization). They span diverse pipelines and metrics (RMSE, log-loss, MAP@K, QWK), including unseen ones. Training tasks are simpler, while held-out tasks are more complex (Appendix §B.2).
>
> **Beyond the benchmark family.**
> We additionally evaluate on 5 held-out tasks from MLE-Bench and Kaggle covering unseen modalities (medical images, audio, time series, multimodal). As shown in **Table R1.1**, **ML-Agent generalizes across these domains**, outperforming same-scale baselines and remaining competitive with much larger LLM-based agents, demonstrating transfer beyond the benchmark family.
>
> **Table R1.1:** Cross-task generalization over 8 trajectories.
>
> | Task Name                                | Task Type                        | Metric | Qwen2.5-7B-Instruct | DeepSeek-R1 | Gemini-2.5-Pro | GPT-5     | ML-Agent  |
> | ---------------------------------------- | -------------------------------- | ------ | ------------------- | ----------- | -------------- | --------- | --------- |
> | APTOS 2019 Blindness Detection           | **medical image** classification | QWK    | -1.10               | -0.61       | -1.51          | -3.35     | **-0.05** |
> | H&M Personalized Fashion Recommendations | **multi-modal** regression       | MAP@12 | -4.26               | -1.06       | 2.13           | **20.21** | **8.51**  |
> | Optiver – Trading at the Close           | **time series** regression       | MAE    | 0.02                | 0.038       | 0.06           | **0.14**  | **0.11**  |
> | ICML 2013 Whale Challenge                | **audio** classification         | AUC    | 0.11                | 0.18        | **5.88**       | 0.57      | **0.72**  |
> | Text Normalization Challenge             | text **normalization**           | Acc.   | -2.49               | 0.08        | **0.13**       | **0.12**  | 0.08      |
>
> ---
>
> ### **W2: Taxonomy of error types**
>
> We agree the taxonomy should be more explicit.
>
> * $F_{\text{error}}$: **agent-caused failures** (e.g., syntax errors), penalized.
> * $F_{\text{corner}}$: **resource-induced failures** (e.g., OOM, timeout), not fully evaluable.
>
> We use a coarse design: errors = -1, corner cases = 0, enforcing executability while avoiding bias toward conservative configurations.
>
> Corner cases are rare, and ablations show minimal impact. We will add definitions and examples.
>
> ---
>
> ### **W3: Step-wise RL and expert state dependency**
>
> We agree step-wise RL may depend on expert states, but results suggest the policy is not restricted:
>
> * States include full trajectory history, including failures.
> * Expert and SFT perform poorly, while RL improves +16.4%.
> * Behavioral analysis (Table R3.2): fewer errors (-55%), more backtracking (+17%), more composite strategies (>2×).
>
> Extreme OOD recovery remains challenging and will be clarified.
>
> ---
>
> ### **Q1: What does the SFT stage improve?**
>
> **(1) Action diversity.**
> More diverse actions (Figure 6).
> **Table R1.2:**
>
> | Methods             | Unique Action Ratio |
> | ------------------- | ------------------- |
> | Qwen2.5-7B-Instruct | 0.9226              |
> | Standard SFT        | 0.7856              |
> | ML-Agent-SFT        | 0.9857              |
>
> **(2) Generalization in RL.**
> Standard SFT: +13% held-in, **-12% held-out**.
> Ours: +18% held-in, **+16% held-out**.
>
> **(3) RL stability.**
> RL fails with base/distill models, but succeeds with our SFT.
>
> ---
>
> ### **L1: Failure analysis**
>
> Three failure modes:
> (1) non-executable edits,
> (2) ineffective valid changes,
> (3) resource-limited runs.
>
> Performance correlates with difficulty (r=0.903) and headroom (r=0.843).
>
> Long-horizon failures often stem from trajectories exceeding context limits, indicating limitations in both planning and context management.

---

> > ### Author Rebuttal · Reviewer_iha9 · 2026-04-04
> >
> > My concerns have been addressed, and I keep my original score.

---

> > > ### Author Response · Authors · 2026-04-07
> > >
> > > Dear Reviewer iha9,
> > >
> > > Thank you very much for your time and thoughtful comments. We truly appreciate your constructive feedback and your positive response to our rebuttal.
> > >
> > > Best regards,
> > >
> > > Authors

---

### Decision · Program_Chairs · 2026-04-30

**Decision:**

Accept (regular)

**Comment:**

This paper presents ML-Agent, a framework for training LLM-based agents to autonomously perform machine learning engineering tasks on Kaggle competitions. The key contribution is a step-wise reinforcement learning approach that enables efficient training of a 7B parameter model to achieve competitive performance with much larger proprietary agents while being 10-100x cheaper to operate.

Most questions have been solved during rebuttal and follow-up discussion stage. Several weaknesses are partially addressed during rebuttal, and I encouraged the authors to further addressed them in the updated version: 1) Limited generalization evaluation beyond related benchmark ecosystems. Rebuttal provides concrete evidence through Table R1.1, but generalization to truly diverse domains beyond these specific modalities remains unvalidated. 2) Missing empirical comparisons with learning-based code approaches. While authors provide conceptual distinction and some empirical grounding, the absence of direct empirical comparison still limits ability to rigorously assess advantage over learning-based prior work. 3) Context limit as fundamental bottleneck. Long-horizon failures often stem from trajectories exceeding context limits, indicating limitations in both planning and context management.

All reviewers are positive to acceptance thus I recommend acceptance.